# Effect of intramedullary nail stiffness on load-sharing in tibiotalocalcaneal arthrodesis: A patient-specific finite element study

Patrick Terrill[1], Ravi Patel [1], Douglas Pacaccio[2], Kenneth Dupont[3], David Safranski[3], Christopher Yakacki[1], Dana Carpenter[1]*

1 Smart Materials and Biomechanics Laboratory, Department of Mechanical Engineering, University of Colorado Denver, Denver, Colorado, United States of America, 2 Advanced Foot and Ankle Surgeons Incorporated, Yorkville, Illinois, United States of America, 3 Clinical Affairs, Foot & Ankle, Enovis, Atlanta, Georgia, United States of America

* dana.carpenter@ucdenver.edu

**Data Availability Statement:** All relevant data are within the paper and its Supporting information files.

## Abstract

Tibiotalocalcaneal (TTC) arthrodesis is a procedure to treat severe ankle and subtalar arthropathy by providing pain free and stable fusion using IM nails. These nails can be manufactured with multiple materials and some feature the ability to dynamize the arthrodesis construct. However, the impact of IM nail material and nail dynamization on load-sharing and in the setting of bone resorption have not been quantified. This work utilized a patient-specific finite element analysis model of TTC arthrodesis to investigate IM nails with differing material moduli and the impact of nail dynamization on load-sharing and intersegmental compression in the setting of bone resorption. Each nail was virtually inserted into a patient-specific model of a hindfoot, which was segmented into the three bones of the TTC complex and assigned material properties based on the densitometry of the bone. Compression, amount of load-sharing, and stress distributions after simulated bone resorption were quantified and compared between the varying IM nails. Simulations revealed that bone segments were only subjected to 17% and 22% of dynamic gait forces in the titanium and carbon fiber nail constructs, whereas the pseudoelastic NiTi nail constructs allowed for 67% of the same. The titanium and carbon fiber nails lost all initial compression in less than 0.13mm of bone resorption, whereas the NiTi nail maintained compression through all simulated values of bone resorption. These data highlight the poor load-sharing of static nail TTC arthrodesis constructs and the ability of a pseudoelastic IM nail construct to maintain intersegmental compression when challenged with bone resorption.

## Introduction

Tibiotalocalcaneal (TTC) arthrodesis is a surgical salvage procedure performed to treat ankle and subtalar pain and trauma related to various pathologies and diseases of the hindfoot and ankle [1, 2]. The purpose of the procedure is to provide a stable, pain-free union of the bones around the ankle and subtalar joints. Reviews of published clinical literature have documented

**Funding:** The author(s) received no specific funding for this work.

**Competing interests:** I have read the journal's policy and the authors of this manuscript have the following competing interests: Safranski and Dupont are paid employees of Enovis Foot & Ankle. Pacaccio is a paid consultant/advisor to Enovis Foot & Ankle. Safranski reports stock ownership and other compensation from MedShape-acquired by DJO during the conduct of this study and outside the submitted work. Dupont reports stock ownership and other compensation from MedShape-acquired by DJO during the conduct of this study and outside the submitted work. This does not alter our adherence to PLOS ONE policies on sharing data and materials.

TTC arthrodesis union rates as high as 94.6%; however, the occurrence of non-union can be seen in up to 50% of procedures involving complex cases of ankle disease (e.g., revision surgeries, diabetes, Charcot neuroarthropathy, tobacco usage, bulk bone defect, etc.) [3–7]. For these challenging cases, newer devices that enable dynamization of the arthrodesis construct (intermittent compressive loading transferred to the bone via elongated screw slots or other sliding mechanisms) may provide for better fusion outcomes.

As described by AO/ASIF principles, an arthrodesis device must provide compression as well as stability to promote successful union. Intersegmental compression helps induce construct stability while promoting union by maximizing bone-to-bone contact across the fusion site. Compression aids in promoting primary bone growth by preventing excessive micromotion of the joint, which is necessary for proper fusion. Additionally, the device must provide rigidity to the fusion site to prevent excessive bending and torsional motions of the TTC complex [8–10]. The application of adequate compression and stability must be accomplished while ensuring adequate load-sharing between the arthrodesis device and bone tissues as dictated by Wolff's Law [11], as excessive strains/micro-motion can result in the formation of fibrous tissue, which prohibits fusion success [12, 13]. Because sustained compression is advantageous for achieving proper fusion, localized bone resorption at the fusion site is of concern. Studies evaluating the clinical performance of pseudoelastic intramedullary (IM) nails have reported average radiographically measured values of resorption-induced shortening ranging from 3.1 mm to 5.6 mm [4, 6, 14, 15]. This degree of shortening would be expected to result in a decrease in compression across the TTC complex; however, the extent of the loss of compression has not been quantified.

In efforts to quantify the impact of arthrodesis devices on TTC construct biomechanics, previous biomechanical studies and development efforts have primarily focused on the stiffness of the TTC-nail construct [2, 16–20] or compression generated from the installation of the IM nail [21–23]. However, to the best of the authors' knowledge, no studies have attempted to quantify how IM nails affect load-sharing between the arthrodesis device and native bone across the fusion site. While some reports have claimed IM nails create a load-sharing construct, these same reports have shown broken locking screws, suggesting that load-sharing in the construct is not controlled [24]. Other IM nail designs allow for dynamization and load-sharing via a second surgery to remove a locking screw from the nail, but this adds additional recovery time, cost, patient pain, and inconvenience [25]. Recent studies have reported positive clinical outcomes using newer pseudoelastic nails (e.g., DynaNail™, Enovis) which utilize a nickel-titanium (NiTi) compressive rod to apply sustained dynamic compression across the arthrodesis site [5, 14, 15, 26]. However, there still lacks a systematic investigation that quantifies the impact of nail stiffness and dynamization on load-sharing within the TTC arthrodesis complex.

To address this knowledge gap, a patient-specific finite element modeling technique was utilized to investigate the impact of IM nail stiffness and dynamization on TTC complex compression, load-sharing, and structural properties in the as-implanted state and in the setting of bone resorption. We hypothesized that dynamization would allow for 1) improved load-sharing and 2) sustained intersegmental compression in the presence of localized bone resorption at the fusion site.

## Materials and methods

### Intramedullary nails

Two different IM nails were modeled for simulation in this study (Fig 1). First, a 10 mm diameter VersaNail® (Zimmer Biomet, Warsaw, IN, USA) was modeled to represent static IM nails

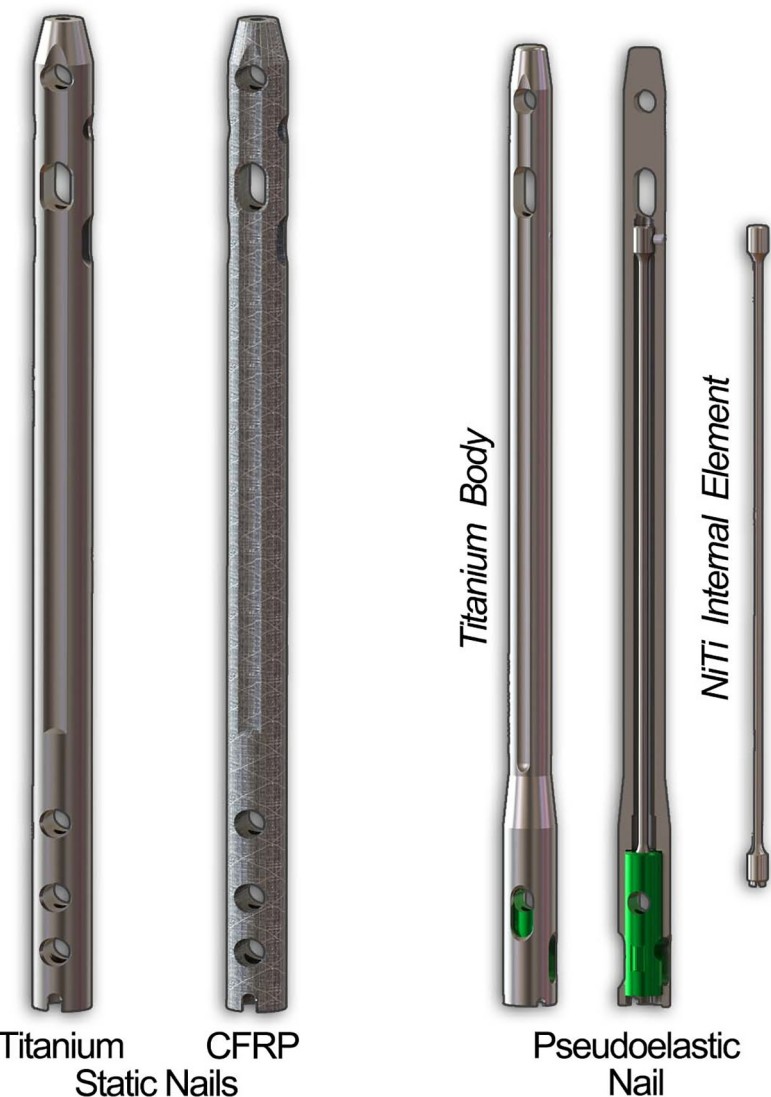

**Fig 1. Renderings of the three IM nails used in this study.** The static nails were modeled after a 10 mm VersaNail using both titanium and CFRP material properties, while the DynaNail represented a dynamic nail with an internal NiTi element inside a titanium outer body.

made from both titanium and carbon-fiber reinforced polyetheretherketone (CFRP). While VersaNail is not currently manufactured in CFRP, this allowed us to create a direct comparison of the effect of material properties on performance. Next, a 10 mm diameter DynaNail® (MedShape-acquired by DJO Global, Atlanta, GA, USA) was modeled to represent a pseudoelastic NiTi nail. In contrast to IM nails that use external rods or internal screws to generate compression, the pseudoelastic NiTi nail uses an internal NiTi element that is stretched across the joints to be fused and partially released to generate compression [27].

## Patient-specific model generation

The right leg from a 55-year old, 82-kg male donor was obtained from Science Care, a non-transplant tissue bank accredited by the American Association of Tissue Banks. All Science

Care donors are fully educated on the tissue and organ donation process, and signed consent is obtained in all cases. Because this study did not involve living human subjects, the Colorado Multiple Institutional Review Board (COMIRB) did not require committee approval of study protocols. A copy of the COMIRB decision tree for defining human subjects research is included in the S1 File. Quantitative computed tomography (QCT) images of the leg were taken and utilized to build a patient-specific finite element model (Fig 2). The specimen falls within the average age range for patients requiring hindfoot fusion (typically between 50 and 62 years) [28, 29]. Images were acquired using a Phillips Gemini 64 slice scanner (Phillips, Amsterdam, The Netherlands) at slice thickness of 0.67 mm and 120 kVp. Ankle joint positioning was maintained in neutral dorsiflexion-plantarflexion and neutral varus-valgus according to the suggested surgical techniques for IM fusion. Images were segmented into the individual bones and meshed for FEA using Simpleware ScanIP (Synopsys, Mountain View, CA, USA). Using a reference phantom (QCT Pro, Mindways, Austin, TX, USA), each voxel in the scan was converted to bone mineral density. Next, using the relationships described by Keyak *et al.*, an elastic modulus for each voxel was defined [30]. The overall stiffness of the

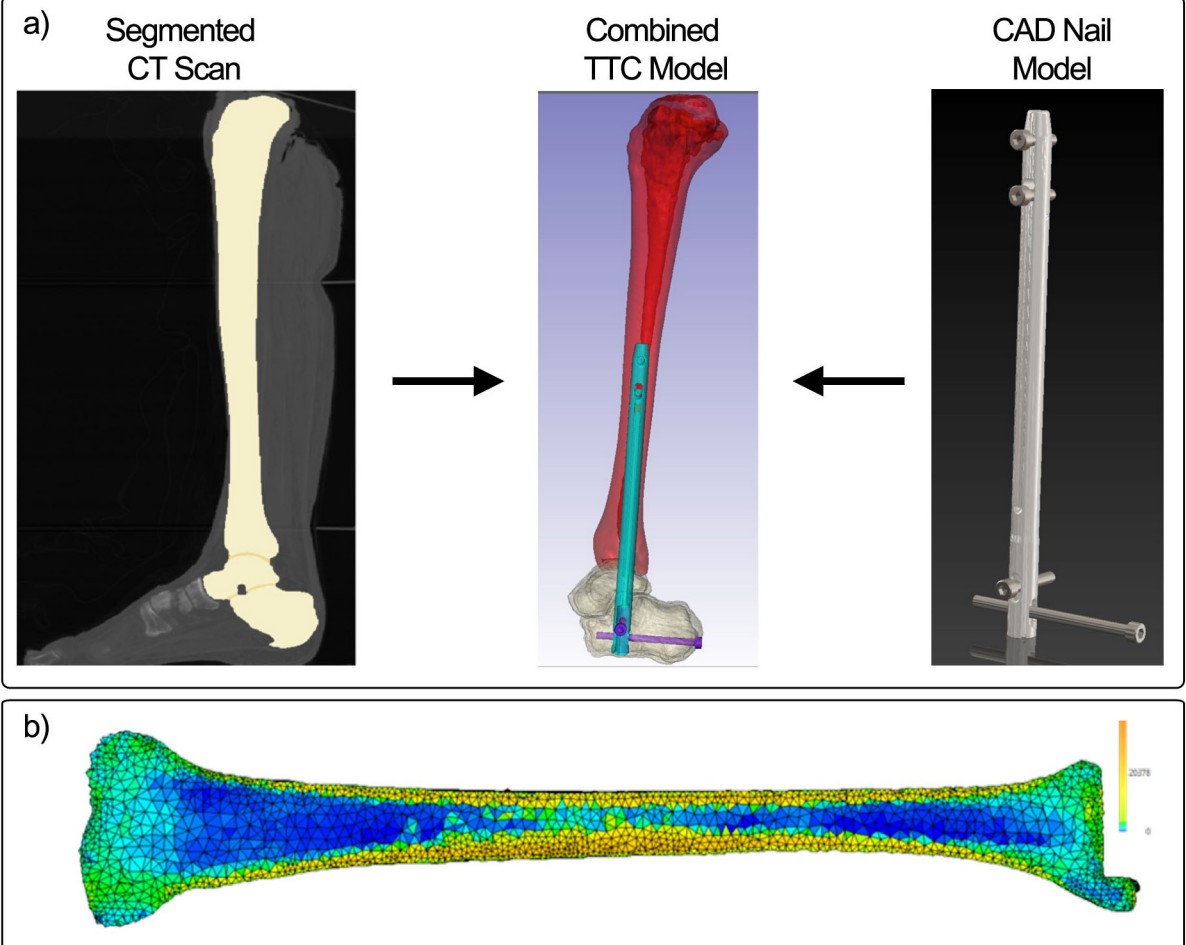

**Fig 2. Illustration of process for patient-specific finite element modeling.** A) Segmented CT images are combined with a CAD model of an IM nail to model an implanted device for TTC fusion. B) The densitometric properties of the bone are converted to modulus values to capture the distribution of properties in the bone allowing for the same patient-specific model to be applied in all 3 nailing systems.

TTC bones was calculated in this study by applying compression across the nail and measuring the force required to reduce the TTC construct length by 1 mm.

The static IM nail was modeled after a 10 mm VersaNail using SolidWorks (Dassault Systems, Waltham, MA, USA), whereas the design files for the pseudoelastic nail were provided by the manufacturer. Material properties for the nail bodies were defined based on well-known material properties; for example, Ti6Al4V and CFRP were assigned modulus values of 110 GPa [31] and 13 GPa [32–34], respectively, to match the materials used in commercially available nails. All fixation screws for the nails utilized the titanium material properties. Each nail was inserted into the model using the ScanIP (Simpleware) software. Nails were oriented to reflect the documentation that is available from the respective manufacturers. The internal NiTi element in the pseudoelastic nail was assigned material properties using the superelastic material model developed by Auricchio et al. and further refined by Anderson et al [31, 35]. Contact surfaces (coefficient of friction = 0.1) were defined between the outer nail body and surrounding bone and at the bone-bone contact regions in the ankle, and the outer surfaces of all screws were bonded with the surrounding bone tissue [36]. Frictionless contact was assigned to the sliding element in the pseudoelastic nail to allow it to slide freely within the nail body. Finally, models were meshed with linear tetrahedral elements and imported into ABAQUS (version 6.14, Simulia, Dassault Systems, Waltham, MA, USA). Mesh convergence and validation via comparison with experimental measurements of bone-IM nail construct stiffness were reported previously [31]. In brief, the mesh was refined until the computed stiffness of the TTC complex under uniaxial compression changed by less than 2%, and the final model consisted of a total of approximately 1.8 million tetrahedral elements. The resulting model had an axial stiffness within 5% of that for a cadaveric TTC complex previously measured by our research team (model stiffness = 2696 N/mm; cadaver stiffness = 2574 N/mm) [23].

## Compression generation

Compression across the fusion interface was generated before analyzing the models for load-sharing or simulated resorption behavior. For the static titanium and CFRP nails, in clinical practice a set screw is used to achieve compression across the fusion site at the time of surgery. To apply a comparable level of compression in the models, uniaxial thermal contraction of the nail body in the axial direction was applied to reach 500 N of compression. To do so, a thermal coefficient of expansion was assigned to the material in the longitudinal direction of the nail, while the coefficients of expansion in the other two orthotropic directions (and to all other materials in the models) were set to zero. Temperature boundary conditions were then applied to cool the nails by 1˚C, causing contraction in the longitudinal direction and generation of the desired 500-N force across the fusion site. Thermal expansion coefficients were adjusted so that the temperature decrease of 1˚C shortened the length of the titanium and CFRP nails by 0.04 and 0.088%, respectively. The 500-N compression achieved by this method was comparable to the compression achieved in clinical applications of other IM nails [27]. Compression was applied to the pseudoelastic nail model following the methods described by Anderson et al [31]. In brief, the compressive element was initially stretched by 8 mm and then relaxed by 2 mm of displacement, after which it was locked in place to the nail body and sliding component within the nail yielding approximately 400 N of compression across the bones. 400 N of compression was the maximum amount of compression which could be applied due to the cross-sectional area of the NiTi element within the nail. This limitation is due to the nature of the pseudoelasticity of NiTi metal applying constant stress over a range of strains. After compression was applied, either gait loading or resorption of the bone was applied to each model to

evaluate load-sharing and sustained compression, respectively. Additional details on the methodology used can be found in the S2 File.

## Load-sharing analysis

A peak compressive load of 1,121 N was applied to the modeled TTC-nail construct to evaluate load-sharing within the ankle-hindfoot to match maximum vertical ground reaction forces measured during walking. These measurements were provided by the Neuromuscular Physiology Lab at The Georgia Institute of Technology [31]. The load was divided among 4 nodes equally, 280.25 N per node, at the distal end of the calcaneus surrounding the opening where the IM nail was implanted (Fig 3). The applied load was also applied exclusively in the z-direction as it is the most substantial component of the gait load during walking. Nodes on the tibial plateau were held fixed. To analyze load-sharing, the model was sliced perpendicular to the axis of the nail at level of the distal tibial screw. Stress normal to the plane of the cut was integrated across each of the different components, and the resultant forces for the nail body, bone, and internal NiTi element (if applicable) were recorded. Load-sharing percentages were then calculated as the resultant force of the bone divided by the applied compressive load.

## Simulated resorption analysis

While it is recognized that bone resorption will occur across the fusion site, the exact amount of bone resorption that will occur in any patient is unknown. Pelton *et al.* reported at least 0.5

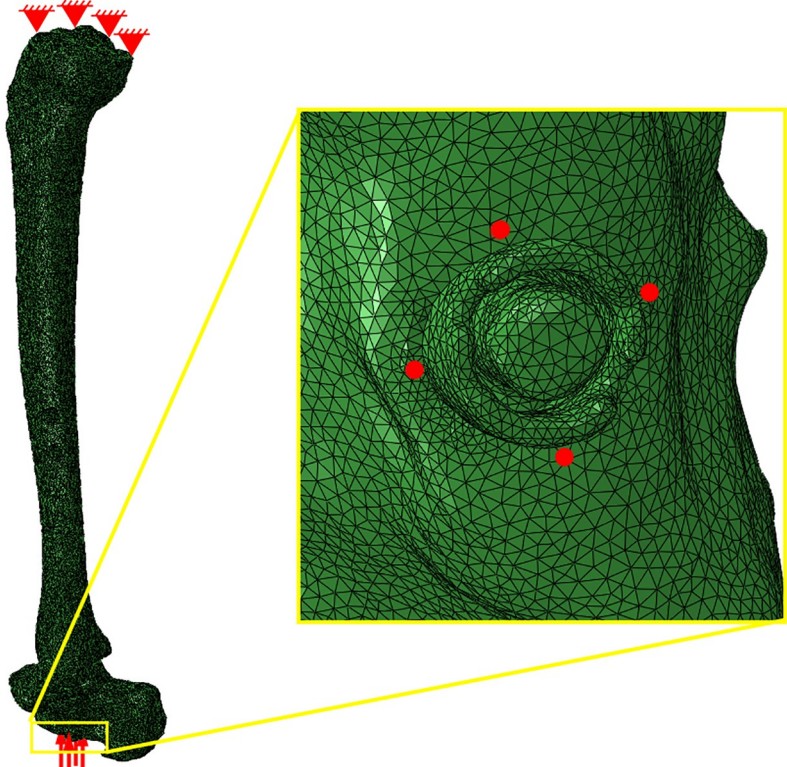

**Fig 3. Boundary conditions applied for load sharing analysis.** Nodes on the proximal surface of the tibial plateau were pinned, and the total gait force was divided among four nodes (indicated by the four red dots in the yellow callout box) on the distal surface of the calcaneus surrounding the distal end of the IM nail.

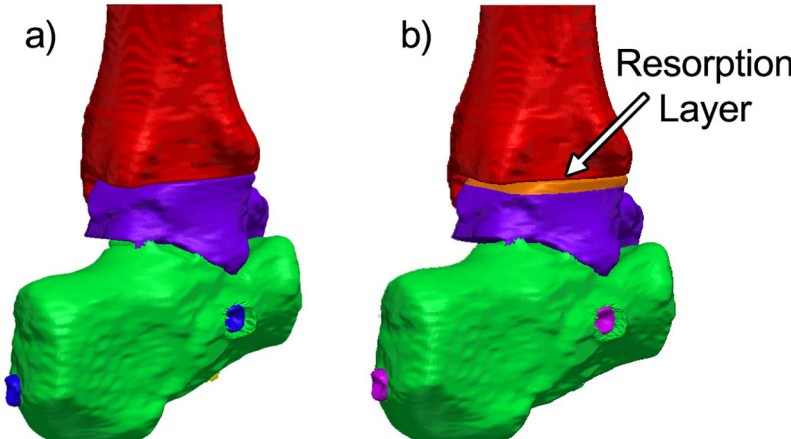

**Fig 4. Load sharing and resorption models.** Model of the TTC complex used for load-sharing analysis. Each zone of different color represents a model component that can be assigned specific material properties and mechanical behavior. **B)** A 3.2 mm portion of the talus was designated to contract to simulate bone resorption.

mm of resorption in 28 patients treated with the VersaNail [37]. Previous case studies using the DynaNail show radiographic evidence suggesting that several millimeters of resorption can occur [4–6, 14, 15]. Therefore, in this study, we systematically simulated a range of values to understand how the devices would behave with increasing amounts of resorption. Resorption was applied by a uniaxial thermal contraction of the resorption zone, using the methods described previously, in the superior-inferior direction (Fig 4). Approximately 0.5 mm of resorption was applied in all three models, and forces through the bone and nail were measured as a function of varying levels of resorption. The percentage of sustained compression was calculated by dividing the resultant compressive force in the bone after resorption by the initial compressive force applied to the model before resorption.

## Results and discussion

### Load-sharing

The first half of this study analyzed the load-sharing behavior of three different types of IM nails. Stress maps were produced for the unloaded state and at peak compressive load to illustrate how stress is distributed across the TTC complex under simulated walking conditions (Fig 5). In these images, the IM nail was hidden from the analysis such that the variation in stress in the bone is observed more clearly. For the static titanium nail, the initial compression of 500 N in the nail generates stress across the TTC complex. When focusing on the talus and distal portion of the tibia, typical stresses are within the range of 1 to 2.5 MPa. As peak loading is reached, there is an increase in stress throughout the joint, as indicated by a larger area of bone converting from blue (~0 to 1 MPa) to green (~1.5 to 3 MPa). In comparison, the pseudoelastic NiTi nail starts with a lower typical average stress across the talus and tibia. This is due to the initial compressive force value being lower than the static titanium nail (i.e., 400 N vs. 500 N). A similar response can be seen during peak loading, with an increase of stress distributed across the joints when loaded. Further, stresses below the subtalar joint and in the calcaneus at peak load were found to be higher in the pseudoelastic nail ($\approx$ 2.3 MPa) compared to the titanium nail ($\approx$ 0.8 MPa), indicating enhanced load-sharing with the bone.

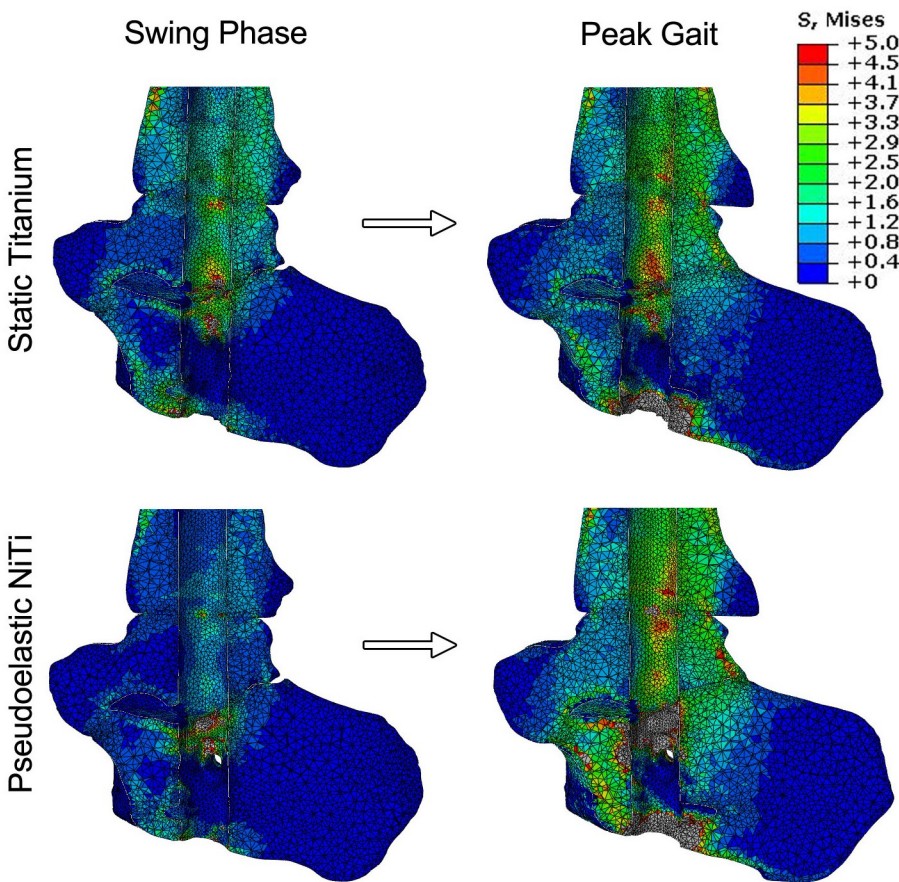

**Fig 5. Bone stresses under simulated gait loading.** Stress maps showing the distribution of stress within the ankle at swing phase and peak loading. In these images, the IM nail was subtracted from the analysis to show the stress changes in the bone only. The static CFRP nail is not shown due to its close similarity to the static titanium nail.

Two cycles of compressive loading, approximating the vertical component of the ground reaction force during gait, were applied quasistatically to each TTC model at discrete time points (44 for the static nails and 64 for the pseudoelastic nail), and the forces through the bone and nail body were recorded (Fig 6A). For each nail, the loading behavior was consistent between both cycles. Due to the different behavior of NiTi during lengthening and shortening, two cycles were simulated to account for any residual deformation remaining at the end of the first cycle. When analyzing load-sharing in the ankle and hindfoot, it is important to note that at the start of the gait cycle (*i.e.* swing phase). When the applied force is 0 N, the magnitude of forces within the bone and devices are equal to the initial value of applied compression; however, these forces are equal and opposite as the device is in tension (+) while the bone is under compression (-). Load-sharing analysis is primarily focused on how these forces change dynamically when an external load of 1,121 N is applied. The static titanium nail (Fig 6B) showed relatively little force variation through the bone ($\Delta F_{bone}$) with the majority of load being transferred through the nail jacket. The nail body experienced loads ranging from approximately 775 N to -150 N, indicating the nail was transitioning from a state of tension (applying compression to the bone segments) to a state of compression (shielding the bone segments from applied loads). The bone in this method experienced compressive loading ranging from -775 N to -960 N, an amplitude of 195 N (illustrating the

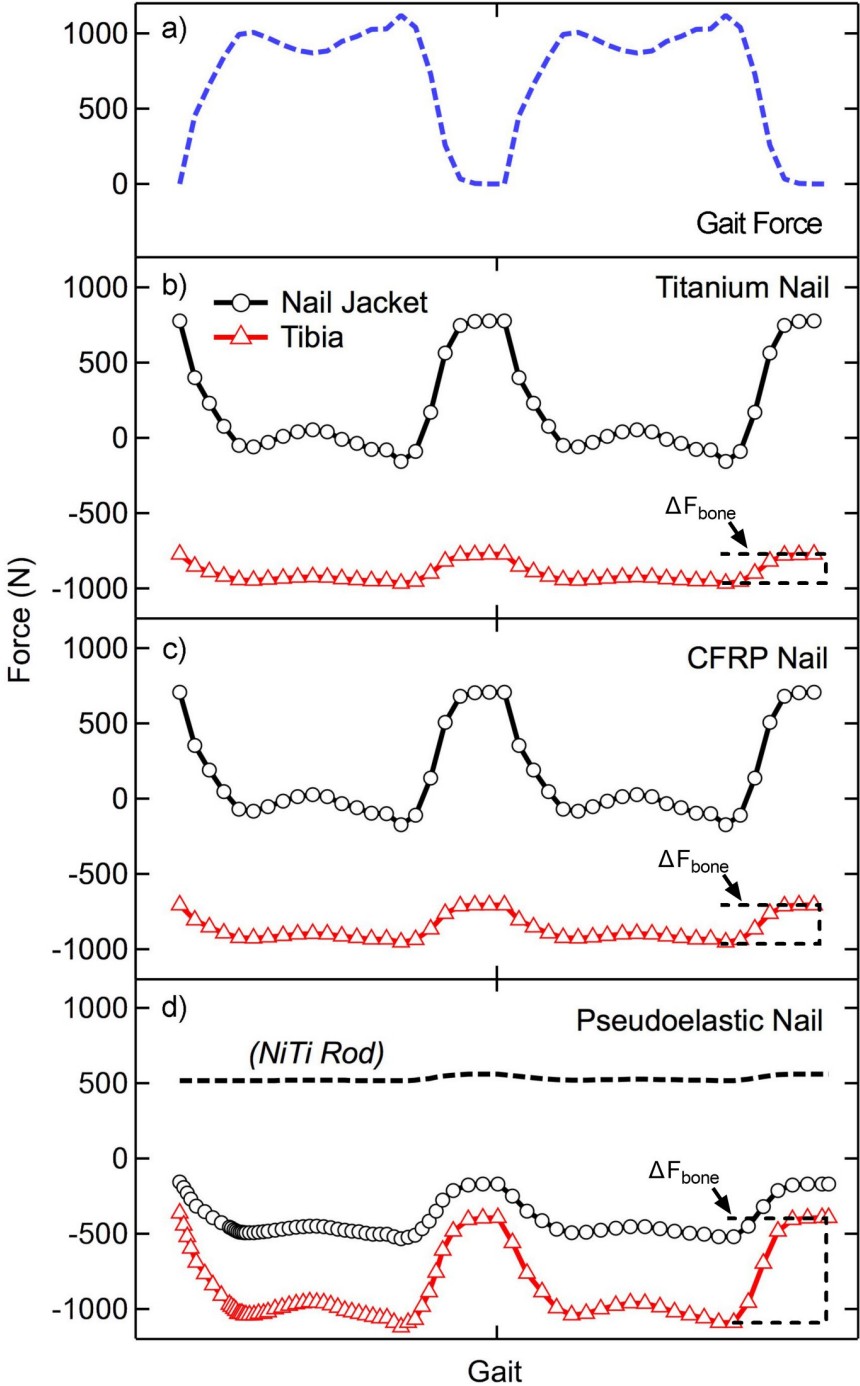

**Fig 6. Applied forces and load sharing results.** A) Two cycles of applied loading used for this study. The change in force as a function of loading for the nail jacket and tibia is shown for (B) static titanium and (C) static CFRP nails. (D) The pseudoelastic nail has an extra component, the NiTi internal element, which is added to the analysis. The change in bone loading is highlighted with brackets for each device type and indicated with $\Delta F_{bone}$.

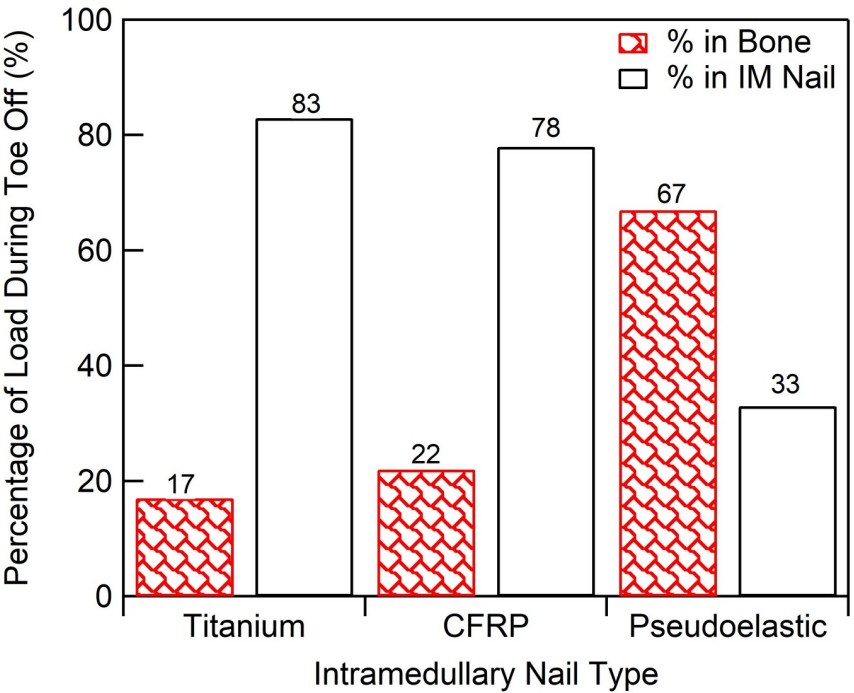

**Fig 7. Load-sharing summary.** Load-sharing percentage of total compressive load transferred to bone for each IM nail.

load bypasses the arthrodesis bone segments through the fixation device). By comparison this is a much smaller load compared to the 925 N amplitude experienced by the nail body. At toe-off, (second peak in applied load shown in Fig 7), 83% of the total load was being transferred through the nail body rather than bone, demonstrating a large amount of stress shielding. Thus, the remaining 17% of the total load was shared to the bone. The static CFRP nail (Fig 6C) showed a similar response to the static titanium nail, with a near identical loading response. At toe-off, the nail body carried 78% of the total load, while the bone experienced 22% of the total load ($\Delta F_{bone}$), thus the nail shielded the bone from the majority of applied load. The pseudoelastic NiTi nail (Fig 6D) was the only nail type which demonstrated significant load transfer to the bone with loads through the tibia ranging from -365 N to -1040 N, a cyclic amplitude of 675 N. The nail jacket loads in this case ranged from -156 N to -490 N with a near constant load of 514 ± 44 N within the NiTi internal element. At toe-off, 33% of the load was transferred through the device with the remaining 67% applied to the bone ($\Delta F_{bone}$, Fig 7).

## Simulated resorption

Simulated resorption was modeled for each condition to investigate how loss of bone at the joint surface would influence the compression across the fusion site. The resorption zone was limited to the talus, as represented in Fig 4B, and allowed for up to 0.5 mm of resorption. This was chosen to minimize modelling complexity as well as computational time. A sagittal view of the stress distributions across the ankle before and after resorption are shown in Fig 8. For the static titanium and CFRP nails, the stress maps reveal that the nail body is under stress to generate compression across the ankle. In comparison, the nail body in the pseudoelastic nail

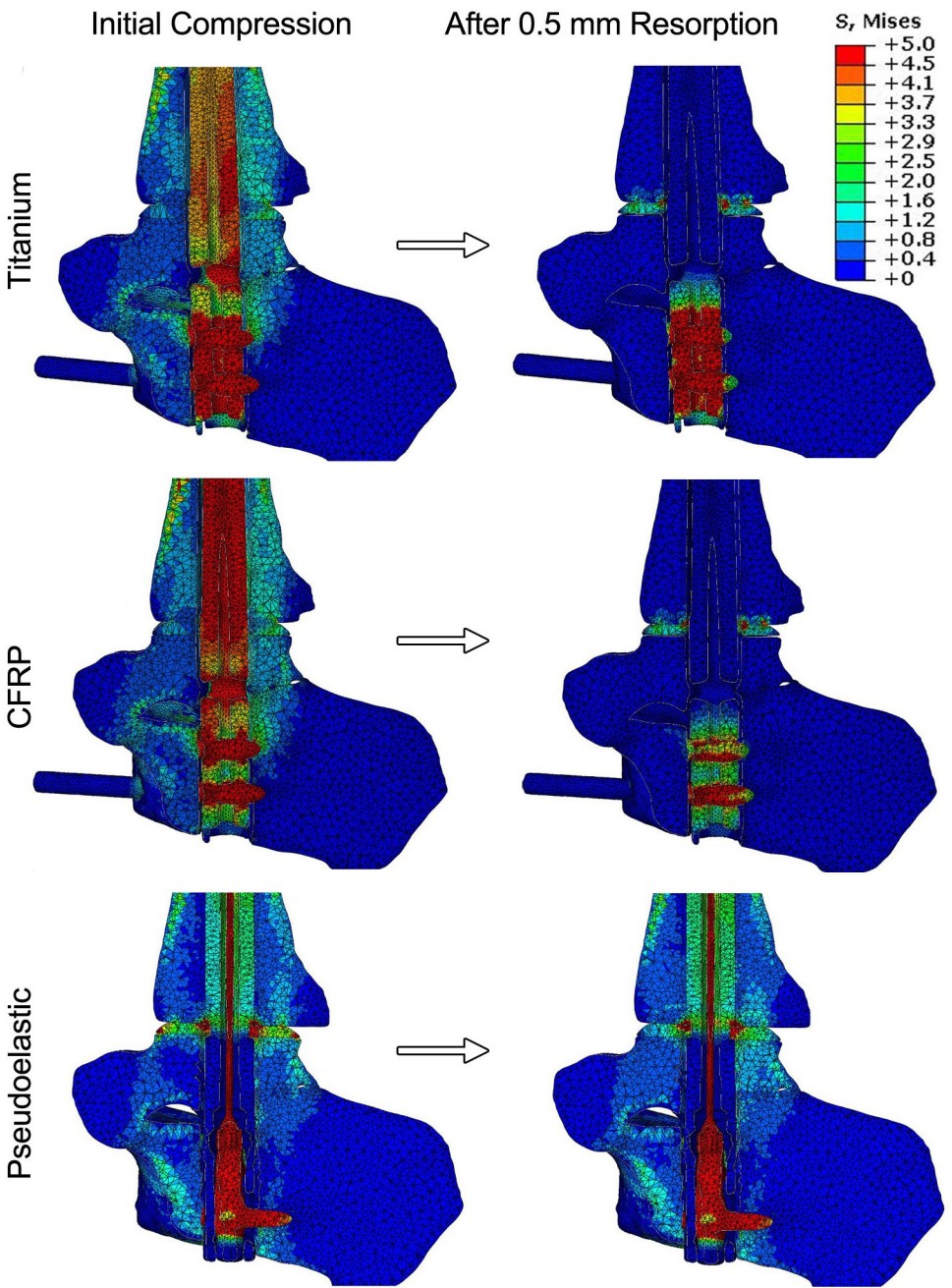

**Fig 8. Results for simulated resorption.** Sagittal slices of the ankle show the stress distribution within the ankle, hindfoot, and device both before and after 0.5 mm simulated resorption. As stresses within the nail bodies and compressive element far exceed those of bone, a limit of 5 MPa was chosen for these maps to allow for adequate differentiation of stresses within the bone.

only experiences peak stresses of approximately 3 MPa; however, the internal NiTi element is responsible for compression generation and experiences well over 5 MPa. For all the nails analyzed, the initial conditions produced stress distributed throughout the joints, with peak values within the range of 2 to 3 MPa. These initial stress values were caused by initial compressions of 500 N for the static nails and 400 N for the pseudoelastic nail.

Simulated resorption values of 0.5 mm were applied to the static titanium, static CFRP, and pseudoelastic nails, respectively (Fig 8). For the static nails, most stress caused by initial compression disappears in the stress map with introduction of resorption, while two distinct regions of elevated stress remained. First, there was a localized region of stress at the proximal portion of the talus, which corresponds to the resorption zone highlighted in Fig 4. This stress results from the thermal contraction (i.e., simulated resorption) of these elements and should therefore be viewed as an artifact of the simulation methodology. This region of elevated thermal stress was limited to the region directly surrounding the nail body (i.e. the location of the view cut displayed in Fig 7), while the mean value for the whole resorption zone consisting of approximately 14,000 elements in each of the three models was negligible (mean = 0.009 MPa, standard deviation = 0.005 MPa). Second, there is a localized region of stress in between the two calcaneal screws in the titanium and CFRP nails. This is a result of the thermal contraction to the nail body designed to induce compression in our model. With the exception of these two localized regions, the stress within the nail and bone is essentially reduced to zero in the CFRP and titanium nails. For the pseudoelastic nail, the distribution of stress throughout the ankle and hindfoot is highly comparable to the initial compression conditions. The stress distributions of bone for each arthrodesis construct before and after bone resorption are provided as violin plots in Fig 9. The width of the plots represents the number of elements that experienced each level of stress, with the median and upper- and lower-quartiles indicated by dashed lines. As seen in these plots, simulated resorption in the static nail models produced a distinct downward shift of the bone stress distribution (i.e., a loss of compression), while simulated resorption in the pseudoelastic NiTi nail model actually produced a slight increase in bone stress. This upward shift in the pseudoelastic NiTi nail model can be attributed to small displacement of the sliding component in the superior direction as the NiTi shortened to maintain compression across the fusion site.

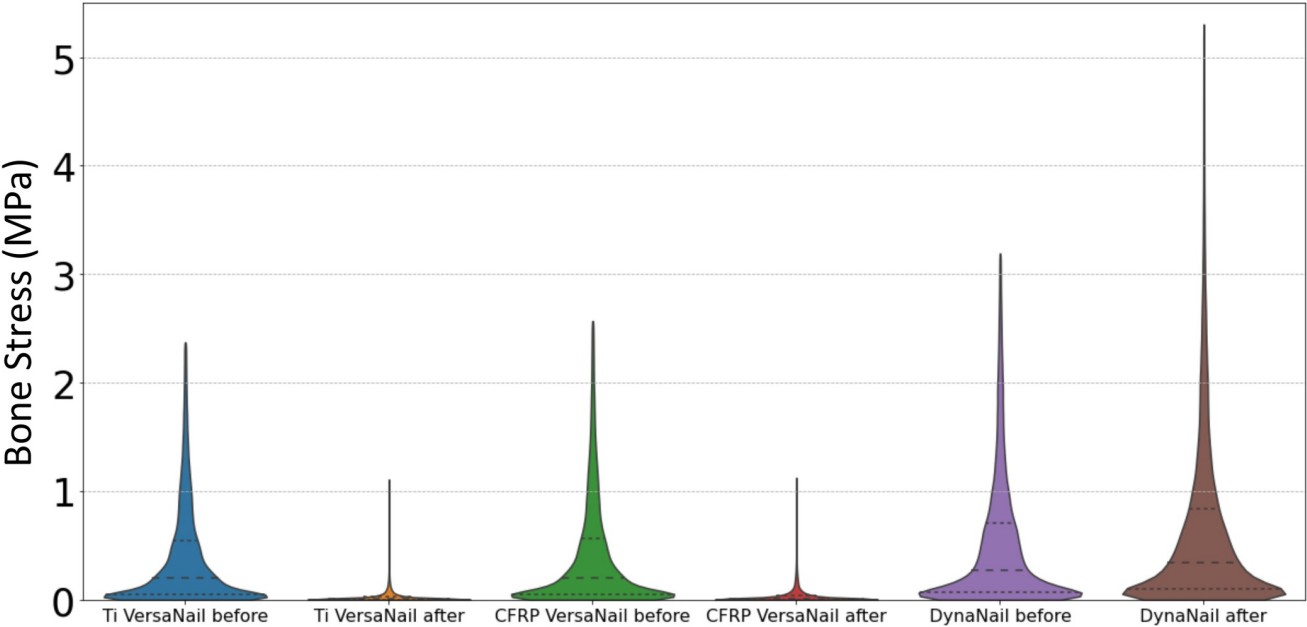

**Fig 9. Bone stress distributions before and after simulated resorption.** Violin plots depict stress distributions in bone for each arthrodesis construct before and after 0.5 mm simulated resorption.

The resultant compression forces within the ankle and hindfoot were calculated to systematically investigate the influence of increasing resorption (Fig 10). For the static titanium and CFRP nails, initial compression was approximately 500 N, but the loads seen during resorption rapidly decreased linearly to 0 N after approximately 0.06 mm of resorption for the titanium nail and approximately 0.13 mm for the CFRP nail. The pseudoelastic nail, however, applied approximately 400 N of initial compression and maintained that loading across the entire resorption range. It should be noted that these models are utilizing an idealized material model for the bone and titanium, as such, the viscoelastic effects of natural bone are not captured.

Adequate sustained and cyclic compressive loading of bone segments provided by fixation hardware (i.e., IM nails) is essential to successful arthrodesis and bone health, but specifics of the loading environment within both hardware and fixed bones, and the way this environment varies with bony resorption changes over time, has previously remained undetermined. These data suggest a minimal degree of compression/load-sharing applied to bone segments in simulated TTC arthrodesis constructs using static compression devices. Specifically, the two static devices, independent of material stiffness, minimally load-shared with the bones, while the pseudoelastic nail had three times the compression/load-sharing capacity ($\Delta F_{bone}$) of the static nails. Furthermore, traditional static nails lost 100% of applied bone compression during simulated bone resorption, whereas the pseudoelastic nail maintained compression through all simulated resorption values (up to 0.5 mm).

Load-sharing data in this study suggests that traditional static nails made from titanium shield a majority of the load from the bone, with approximately 83% of the compressive load being transferred through the nail body, and only 17% being transferred through the bone. Because of this phenomenon, new nail designs have been created while utilizing lower

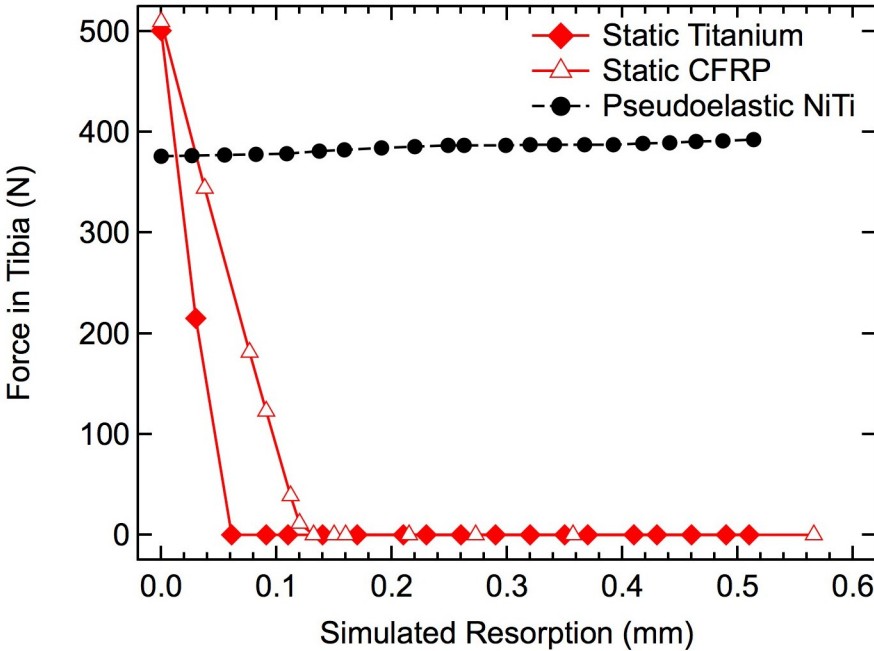

**Fig 10. Effects of simulated resorption on tibial force.** The compressive force transmitted through the tibial diaphysis is shown for the two static IM nails (red) and pseudoelastic IM nail (black) with increasing amounts of resorption.

modulus materials such as CFRP to reduce the overall stress shielding effect. However, this study illustrates that even a CFRP nail with ~10x decrease in modulus still stress-shielded approximately 78% of the load from the bone (a net 5% change in loading). This result is likely due to the high stiffness of the nail rather than the modulus of the material with which the nail is made. IM nails are a construct or structure rather than a material alone, and their mechanical structural properties (such as stiffness) depend on not only the material of which they are made, but also the amount and arrangement of that material (i.e., diameter of nail). For example, the axial stiffness for the titanium nail was approximately 52,000 N/mm while the CFRP nail had a stiffness of 8,200 N/mm, which were 19.3 and 3.0x the stiffness of the TTC bone complex in this study (2,696 N/mm). Resulting from the drastically greater stiffness of the nail in comparison to the TTC bone complex, the nails theoretically should carry proportionately greater load when compressed in parallel, a phenomenon illustrated by the data generated in this work.

In stark contrast, the compressive load applied through the tibia in the pseudoelastic NiTi nail construct was much greater at 67%, with only 33% through the device. This result is achieved by the device's lower axial stiffness from the relatively compliant internal NiTi element (found to be 1,204 N/mm) combined with its already-dynamized screws, thus the enhanced load-sharing as this stiffness is closer to the stiffness of the TTC construct.

The resorption response analyzed in this study also demonstrated that the traditional titanium and newer CFRP nails are unable to provide any sustained compression over resorption ranges even far below those observed clinically [4–6, 14, 15]. The high stiffness of the nails prevents them from maintaining compression when resorption exceeds the elongation of the nail caused by the compressive force in the bone generated during nail insertion. Resulting from the non-uniform geometries and applied bending loads due to gait kinematics in the TTC joint, these data reveal that titanium and CFRP nails are not even capable of reaching their respective theoretical maximum resorption values of 0.4 mm and 0.5 mm [(initial compressive load * nail length)/(nail modulus * cross-sectional area)].

Again, in stark contrast to the static nail designs, these data illustrate the capability of the pseudoelastic nail with the NiTi compressive element to provide compression beyond 0.5 mm of bone resorption. The inherent pseudoelastic properties of NiTi allow for a nearly constant stress within the material as it transitions from the stretched and unstable martensite phase back to the unstretched and stable austenite phase in the crystal structure [38, 39]. This material phase transition behavior is leveraged by the pseudoelastic NiTi nail allowing it to provide sustained compression for 6 mm of contraction (approximately 6% strain), which is an order of magnitude higher than the strain stored in the static IM nails (approximately 0.4% strain) [27, 40]. This behavior enables the pseudoelastic nail device to apply continued intersegmental compression even when challenged with relatively large amounts of bone resorption.

Limitations of this study were that it primarily utilized linear-elastic material properties for the bone, whereas bone is viscoelastic in nature. The effect of this limitation is minimal however, as these viscoelastic effects are typically short lived and are trivial in the case of sustained compression over the course of resorption, which can occur over periods of weeks to months. The bone-implant interface in our models was also idealized. To create our models, each nail was digitally placed in the bone, and the corresponding volume of bone was deleted, producing a cavity in the bone that exactly matched the nail geometry. In clinical application, a reamer is used to create the cavity prior to inserting the nail, producing some surface irregularities that were not captured in our simulations. It should also be noted that the resorption models utilized in this study were limited to approximately 0.5 mm of resorption due to computational instability due to large deformations. However, as shown in the titanium and CFRP nails, compression is lost well within the ranges tested in this study. Another limitation is that the

simulated gait loading used in this study did not include shifts in load location that occur during walking, and it only included the vertical force component, which dominates the ground reaction force magnitude. This loading scenario was intended to capture the overall magnitude and variation of the force transmitted through the fusion site, and it is possible that more complex simulations including shifts of load location and direction would reveal some more subtle shifts in stress levels and locations. Another limitation is that this study only utilized a single, patient-specific FEA model without cadaver testing validation of stress or strain values (only overall stiffness was compared, as noted in the methods section). While physical testing of multiple cadaver specimens can provide valuable information, this approach allowed comparisons between the performance of different IM nails in a single hindfoot model, eliminating the confounding factors of differing bone geometries and bone densities inherent to cadaveric testing. Additionally, this finite element analysis (FEA) technique enabled prediction of stresses throughout the entire bone and device construct, as opposed to point-wise measurements frequently produced during cadaveric testing. In order to fully confirm that the relative nail behaviors analyzed in this study are applicable across a wide range of skeletal anatomies, additional models would need to be developed based on subjects with variable size, age, sex, and bone quality. However, despite this limitation and the others discussed above, the results of this study provide the first steps toward a quantitative understanding of how dynamization and the use of a pseudoelastic IM nail can help provide favorable load sharing conditions and sustained compression in the face of local bone resorption at the ankle.

## Conclusions

In conclusion, a patient-specific finite element model was developed to understand the load-sharing and resorption behavior in 3 different types of intramedullary nails for TTC fusion. It was found that, when digitally implanted into this individual's anatomy, both the titanium and CFRP nails demonstrated poor load-sharing with the bone due to their relatively high stiffness, with approximately 17% and 22% of gait-generated load being transferred to the bone, respectively. The pseudoelastic NiTi nail was found to achieve increased load-sharing, with approximately 67% of the load being transferred through the bone. Additionally, the pseudoelastic NiTi nail maintained substantial compression over the full 0.5 mm of simulated bone resorption while the titanium and CFRP nails both lost all compression within 0.15 mm of resorption. The pseudoelastic NiTi nail modelled in this study represents a marked shift in the design and mechanical performance of IM nails, promoting both load-sharing and compression critical to a successful fusion.

## Supporting information

**S1 File. COMIRB decision tree.** This flow chart is used by the Colorado Multiple Institutional Review Board to determine whether a study falls under the classification of human subjects research. Because cadaveric specimens are not alive, the use of a donated specimen in our study did not qualify as human subjects research.
(PDF)

**S2 File. Ankle fusion FE modeling SOP.** This standard operating procedure from our laboratory provides detailed instructions on how to run finite element analysis with an intramedullary nail digitally implanted in a patient-specific ankle model. Instructions on how to activate the nickel titanium compressive element in a pseudoelastic IM nail are included.
(PDF)

## Acknowledgments

The authors would like to thank MedShape, Inc. (acquired by DJO/Enovis), for providing a pseudoelastic NiTi-based and titanium nail for modeling and Kurt "Burt" Jacobus for his enthusiastic support for the project. The authors thank Vasily Buharin for providing ground reaction force data for gait loading. Safranski and Dupont are paid employees of Enovis Foot & Ankle. Pacaccio is a paid consultant/advisor to Enovis Foot & Ankle.

## Author Contributions

**Conceptualization:** Patrick Terrill, Ravi Patel, Douglas Pacaccio, Kenneth Dupont, David Safranski, Dana Carpenter.

**Data curation:** Patrick Terrill, Ravi Patel, Kenneth Dupont, Christopher Yakacki, Dana Carpenter.

**Formal analysis:** Patrick Terrill, Ravi Patel, Dana Carpenter.

**Funding acquisition:** Douglas Pacaccio, Kenneth Dupont, David Safranski, Dana Carpenter.

**Investigation:** Patrick Terrill, Ravi Patel, Douglas Pacaccio, Kenneth Dupont, Dana Carpenter.

**Methodology:** Patrick Terrill, Ravi Patel, Christopher Yakacki, Dana Carpenter.

**Project administration:** Kenneth Dupont, David Safranski, Christopher Yakacki, Dana Carpenter.

**Resources:** Douglas Pacaccio, Kenneth Dupont, David Safranski, Christopher Yakacki, Dana Carpenter.

**Software:** Kenneth Dupont, David Safranski, Dana Carpenter.

**Supervision:** Douglas Pacaccio, Kenneth Dupont, David Safranski, Christopher Yakacki, Dana Carpenter.

**Validation:** Patrick Terrill, Ravi Patel, Dana Carpenter.

**Visualization:** Patrick Terrill, Ravi Patel, Kenneth Dupont, David Safranski, Christopher Yakacki, Dana Carpenter.

**Writing – original draft:** Patrick Terrill, Ravi Patel, Douglas Pacaccio, Kenneth Dupont, David Safranski, Christopher Yakacki, Dana Carpenter.

**Writing – review & editing:** Patrick Terrill, Ravi Patel, Douglas Pacaccio, Kenneth Dupont, David Safranski, Christopher Yakacki, Dana Carpenter.

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
