## [Decision Letter · Decision Letter 0]

20 Mar 2023

PONE-D-23-03456Effect of Intramedullary Nail Stiffness on Load-sharing in Tibiotalocalcaneal Arthrodesis: A Patient-Specific Finite Element StudyPLOS ONE

Dear Dr. Carpenter,

Thank you for submitting your manuscript to PLOS ONE. After careful consideration, we feel that it has merit but does not fully meet PLOS ONE’s publication criteria as it currently stands. Therefore, we invite you to submit a revised version of the manuscript that addresses the points raised during the review process.

We look forward to receiving your revised manuscript.

Kind regards,

Pawel Klosowski, D.Sc.

Academic Editor

PLOS ONE

Journal Requirements:

"I have read the journal's policy and the authors of this manuscript have the following competing interests: Safranski and Dupont are paid employees of Enovis Foot & Ankle. Pacaccio is a paid consultant/advisor to Enovis Foot & Ankle. Safranski reports stock ownership and other compensation from MedShape-acquired by DJO during the conduct of this study and outside the submitted work. Dupont reports stock ownership and other compensation from MedShape-acquired by DJO during the conduct of this study and outside the submitted work."

Additional Editor Comments:

According to reviewers opinion the paper needs major revision. The details you can read in the opinions. You can submit the corrected version within 45 days.

Reviewers' comments:

Reviewer's Responses to Questions

**Comments to the Author**

1. Is the manuscript technically sound, and do the data support the conclusions?

Reviewer #1: Partly

Reviewer #2: Partly

2. Has the statistical analysis been performed appropriately and rigorously? 

Reviewer #1: N/A

Reviewer #2: N/A

3. Have the authors made all data underlying the findings in their manuscript fully available?

Reviewer #1: No

Reviewer #2: Yes

4. Is the manuscript presented in an intelligible fashion and written in standard English?

Reviewer #1: Yes

Reviewer #2: Yes

5. Review Comments to the Author

Reviewer #1: Dear authors,

I have found this article very interesting, well-written, and well-structured.

I am sending you the following major issues that, by my understanding, should be considered:

1. Regarding the FE models, the information is insufficient for reproducibility (unless the source ABAQUS files are supplied along with the publication).

2. The description of the FEM boundary conditions should be explained in more detail in the article, as in the text and graphically.

4. It is said that the mesh convergence and validation were reported in [31], but this reference does not explicitly show the validation but only comments on it. Could you please extend the explanation?

3. Please explain why linear tetrahedral elements have been used instead of higher-order ones. Also, reference [31] does not specify which element order was used and shows some artificial bending due to using tetrahedral finite elements. Could parabolic elements or finer meshes have fixed the problem? Is this problem also present in the current simulations?

4. The thermal contraction of the resorption zone introduces fictitious (tensile) stresses. How can you analyze the load-sharing behavior when fictitious stresses are involved? Please quantify such stresses and explain why they are neglectable or do not affect the calculation of the resultant forces.

5. Given that the equivalent von Mises stress is signless and more related to shear than normal stresses, why is it the only stress used? Maximum/minimum normal stresses or principal stresses arrow maps would be much more helpful in understanding the system’s mechanics and appropriate for analyzing the stresses in the bone. Moreover, these other stresses allow for distinguishing between tensile and compressive stresses, while von Mises hides this aspect. Also, it can help to quantify and understand the magnitude of the fictitious resorption stresses.

On the other hand, the following minor issues can be considered:

6. A reference for the friction coefficient value of 0.1 between the nail body and the surrounding bone would be necessary.

7. Which friction coefficient value is used for the contact between the nail and the nail body?

8. Why “stress heat map” instead of simply “stress map”?

Sincerely

Reviewer #2: PONE-D-23-03456

Effect of Intramedullary Nail Stiffness on Load-sharing in Tibiotalocalcaneal Arthrodesis: A Patient-Specific Finite Element Study

Terrill et al.

Comments to Authors:

In this manuscript, the authors built a finite element hindfoot model to determine the influence of different intramedullary nails used for tibiotalocalcaneal arthrodesis on the amount of bone load-sharing with and without different bone resorption depths. Overall, the manuscript is well written, but there are some concerns related to the boundary conditions of the model and its ability to appropriately investigate the devices of interest.

Introduction

Lines 48-50, Page 3: A clear and concise explanation of “dynamization of the arthrodesis construct’ should be included prior to or within this sentence to help explain this concept before moving on.

Lines 52-57, Page 3: You state that important factors are maximizing bone-to-bone contact across the fusion site and minimizing micromotion of the joint for proper fusion. You also mention that the device should be rigid to prevent excessive bending and torsional moments at the TTC complex.

How does bone-to-bone contact changes with these different devices?

How does the micromotion change?

Lines 64-66, Page 4: The language should change here to emphasize the comparison of different IM nails here because some of the authors of this manuscript have previously published at least one other study quantifying the load-sharing of arthrodesis devices and bone at the fusion sit, as I understand it.

Lines 78-80, Page 4: For (1), be more specific. The authors are not testing a hypothesis for any material implemented. They are testing a hypothesis for two specific materials. Define ‘high stiffness’. Why not just stick to the second part of the hypothesis? The introduction leads into the second part of the hypothesis, but the first part seems out of place or at least not supported in the Introduction.

Method

Lines 99-100, Page 5: The study is interesting, but the single, healthy subject used to develop the finite element model without experimental validation limits the impact.

Lines 112-113, Page 5: Why not demonstrate the influence of different IM nails with varying material properties, since only one subject was used to develop a single finite element model? In the previous study by the authors, they vary the effect of bone modulus and demonstrate changes. How does the comparison in this manuscript of the different IM nails changes in relation to the bone modulus, and does the positive effect of the dynamization decrease with different bone quality?

Lines 129-130, Page 6: Why not demonstrate the influence from different IM nails with different alignment? How sensitive are the results to changes in position and alignment of the device?

Lines 132-135, Pages 6-7: Where does the coefficient of friction value come from?

Lines 132-135, Pages 6-7: Is the bone-nail interface perfect? Is that realistic? What implications are there for modeling the IM nails this way? This model representation negates the ability of the device to settle with further loading and potentially change the results. This should probably be mentioned in the Discussion.

Lines 135-136, Page 7: Do you mean frictionless here? The language used in this sentence does not make this clear.

Lines 142-143, Page 7: I would refer to this as ‘simulated resorption behavior’ because the representation of resorption is simplified.

Lines 164-166, Page 8: I am confused by the loading applied. How can you apply this load to just the distal end of the calcaneus where the IM nail was implanted and call it ‘gait loading’? What about during toe-off when the ground reaction force is being applied to the forefoot, which wasn’t modeled? Also, why weren’t the off axis loads applied to assess the complexity of loads during stance? The loading does not seem physiologic if all the load is always applied to the distal calcaneus near the IM nail implanted.

Results

Lines 197-198, Page 9: Change ‘no gait loading’ and ‘peak gait load’ to ‘unloaded’ and ‘peak compressive load’. This manuscript may be basing the compressive force off of the ground reaction forces during gait, but they are only being applied as a range of axial compressive loads. They do not replicate the loads under walking conditions, as these would be more complex.

Lines 201-210, Page 10: Why focus on reporting absolute values? Since you only have one subject without any experimental validation, the analysis should consist of relative comparisons between the different nails. There is no ability to determine if these values are correct, or that they represent the values that would be demonstrated in others.

Lines 205-206, Page 10: Why did you compare 400 to 500N? If you can’t achieve 500N with the NiTi nail, then why not just compare each to 400N? I understand that the differences found between the two are vast, even with this discrepancy, but the inconsistency in methodology doesn’t make sense.

Lines 217-224, Pages 10-11: The simulations seem to be quasistatic and not dynamic, so I’m not sure why two cycles of gait loading were performed. It seems like the axial load was just varied. Were dynamic simulations performed and acceleration of loading during walking included?

Lines 232-234, Page 11: It is not clear where at ‘toe-off’ you are referring to make this calculation.

Lines 253-264, Page 12: I like the sensitivity analysis of simulated bone resorption in this study. Why not just focus on this aspect? Since this is one model of a single foot without experimental validation, the sensitivity of different amounts of bone resorption is more interesting and impactful than the earlier analysis.

Discussion

Lines 310-313, Page 14: Can you really say that this is dynamic compressive loading? It is not clear from the methods that you can. It appears you applied different values of compressive load near the site of the nail.

Lines 313-315, Page 14: I would use ‘suggest’ instead of ‘exhibit’ here. The language is a little strong for an analysis with one model.

Lines 369-372, Page 17: Why not include more specimens then to present results in light of different subject variability in bone material properties, geometry, alignment, etc.?

Lines 372-374, Page 17: This is true, but the finite element model would be more impactful with some level of experimental validation and then use this to even extrapolate predictions of stress.

6. PLOS authors have the option to publish the peer review history of their article (what does this mean?). If published, this will include your full peer review and any attached files.

Reviewer #1: No

Reviewer #2: No

---

## [Author Response · Author response to Decision Letter 0]

26 Apr 2023

Response to Reviewers

The authors would like to thank the reviewers for their efforts in thoughtfully evaluating our manuscript. Below we have included each question/comment provided by the reviewers along with our responses. The Revised Manuscript with Track Changes document highlights the changes made in response to the critiques, and the changes are described below. Please note that the line numbers listed in the comments below correspond to the line numbers in the marked up copy. We feel that these changes have resulted in a stronger paper that is now suitable for publication in PLOS ONE. 

Reviewer #1: 

1. Regarding the FE models, the information is insufficient for reproducibility (unless the source ABAQUS files are supplied along with the publication).

The text in the methods section has been edited to include more details about the model, and a new figure (Fig 3) was added to more clearly explain how boundary conditions were applied. In addition, we now include in the supporting information (S2 File. Ankle fusion FE modeling SOP) our lab’s standard operating procedure document with detailed instructions on how to run simulations of the pseudoelastic IM nail in Abaqus. Because this document alone spans 18 pages, our description in the materials and methods section is abbreviated out of necessity. The additional information added to the manuscript and supporting information provides the details needed for reproducibility.

2. The description of the FEM boundary conditions should be explained in more detail in the article, as in the text and graphically.

A new Fig 3 was added to the manuscript to more clearly present the pinned boundary condition at the proximal surface of the tibia and the four force vector locations on the distal surface of the calcaneus.

4. It is said that the mesh convergence and validation were reported in [31], but this reference does not explicitly show the validation but only comments on it. Could you please extend the explanation?

Additional information about mesh convergence (refined until total model stiffness varied by less than 5%) and validation (stiffness of model compared with that of experimental stiffness measurements) has been added to page (lines 149-154).

3. Please explain why linear tetrahedral elements have been used instead of higher-order ones. Also, reference [31] does not specify which element order was used and shows some artificial bending due to using tetrahedral finite elements. Could parabolic elements or finer meshes have fixed the problem? Is this problem also present in the current simulations?

Linear elements were used to help with computational efficiency. The final mesh for the pseudoelastic nail contained 1.8 million elements and accounted for the complex, nonlinear material behavior of nickel titanium. This model pushed the limits of our laboratory’s computational resources, and adding the additional degrees of freedom offered by quadratic elements was deemed unnecessary. The relatively simpler static nail models could use quadratic elements and still be solved in a reasonable amount of time in our lab, but we thought it was more important to maintain a consistent element type in all three models. The gradient in stress noted in reference 31 from 2016 (attributed to a slight bending component produced by the NiTi wire) is now understood to be due to the combination of screw placement and the inherent asymmetry in placement of pin boundary conditions on the tibial plateau, neither of which are expected to be significantly affected by element type. 

4. The thermal contraction of the resorption zone introduces fictitious (tensile) stresses. How can you analyze the load-sharing behavior when fictitious stresses are involved? Please quantify such stresses and explain why they are neglectable or do not affect the calculation of the resultant forces.

The thermal contraction used to simulate bone resorption in our model indeed results in stresses in the resorption zone. Load sharing results under simulated resorption conditions were evaluated in the tibial diaphysis, proximal to the resorption region. Because the distal end of the model is free to move, the stresses in the resorption zone lead to contraction across the fusion site and a drastic reduction in force passing through the tibia for the static nails, as shown in Fig. 10 (Fig. 9 in the original manuscript). The thermal stresses that remain (mean value =0.009 MPa +/- standard deviation of 0.005) in the 14,055 elements of the resorption region) are essentially the same for all three nail models and are negligible compared with the stresses in other locations. We have edited the text describing the resorption results (lines 306-310) to make this clearer.

5. Given that the equivalent von Mises stress is signless and more related to shear than normal stresses, why is it the only stress used? Maximum/minimum normal stresses or principal stresses arrow maps would be much more helpful in understanding the system’s mechanics and appropriate for analyzing the stresses in the bone. Moreover, these other stresses allow for distinguishing between tensile and compressive stresses, while von Mises hides this aspect. Also, it can help to quantify and understand the magnitude of the fictitious resorption stresses.

The reviewer is correct that the use of von Mises stress precludes the ability to differentiate between zones of tensile and compressive stress. We chose to focus on von Mises stress, because it is one way of summarizing the entire stress state at each point with a single value indicating the level of distortional stress. Due to the compressive loads created by the IM nails and the simulated ground reaction forces, stresses in our models are by far dominated by compression (other than the tensile stress in the nickel titanium component of the pseudoelastic nail). There are no areas of notable tensile stress, other than the thermal stresses in the resorption region which, as noted in the previous comment, are negligible compared with stresses in the bones and nails.

6. A reference for the friction coefficient value of 0.1 between the nail body and the surrounding bone would be necessary.

The following literature source is now cited: Yu HY, Cai ZB, Zhou ZR, Zhu MH. Fretting behavior of cortical bone against titanium and its alloy. Wear 2005;259:910-918. This study found the friction coefficients in the range of 0.17-0.29 for small displacements between titanium and cortical bone. During model development, we tested coefficients ranging from 0 (frictionless) to 0.4 and found no noticeable difference in the resulting stress distribution.

7. Which friction coefficient value is used for the contact between the nail and the nail body?

All metal components (screws and nail bodies) were treated as bonded (nodes shared at the interface), except for the interface between the sliding element and body in the pseudoelastic nail, which was modeled as frictionless contact.

8. Why “stress heat map” instead of simply “stress map”?

This was simply a style choice of the authors. We have removed “heat” from all mentions of the stress maps in the paper.

Reviewer #2:

Introduction

Lines 48-50, Page 3: A clear and concise explanation of “dynamization of the arthrodesis construct’ should be included prior to or within this sentence to help explain this concept before moving on.

A brief statement describing dynamization has been added to this paragraph.

Lines 52-57, Page 3: You state that important factors are maximizing bone-to-bone contact across the fusion site and minimizing micromotion of the joint for proper fusion. You also mention that the device should be rigid to prevent excessive bending and torsional moments at the TTC complex. How does bone-to-bone contact changes with these different devices? How does the micromotion change?

Bone-to-bone contact is identical in all three of our models, because the same bony geometry was used for each. We did not explicitly measure micromotion. The models provide a huge amount of data on different mechanical quantities, including stress and strain (individual stress/strain components, principal stresses/strains, pressure, deviatoric stress, and other summary values like von Mises stress, octahedral shear stress, strain energy density, etc.), displacements, and reaction forces, among others. In order to keep our analysis focused and succinct, we chose to focus on stress (von Mises stress specifically) and load sharing, as we feel these two quantities best summarize the mechanical aspects we aim to understand (load sharing between the devices and bone and influence of bone resorption on stress generated across the fusion site).

Lines 64-66, Page 4: The language should change here to emphasize the comparison of different IM nails here because some of the authors of this manuscript have previously published at least one other study quantifying the load-sharing of arthrodesis devices and bone at the fusion site, as I understand it.

We have revised this sentence to state, “…however, to the best of the authors’ knowledge, no studies have attempted to quantify how different IM nail designs and materials affect load-sharing between the arthrodesis device and native bone across the fusion site.”

Lines 78-80, Page 4: For (1), be more specific. The authors are not testing a hypothesis for any material implemented. They are testing a hypothesis for two specific materials. Define ‘high stiffness’. Why not just stick to the second part of the hypothesis? The introduction leads into the second part of the hypothesis, but the first part seems out of place or at least not supported in the Introduction.

Thank you for this suggestion. We revised our hypothesis statement to read, “We hypothesized that dynamization would allow for 1) improved load-sharing and 2) sustained intersegmental compression in the presence of localized bone resorption at the fusion site.” The new statement provides more specific information and more accurately reflects the purpose of the study.

Methods

Lines 99-100, Page 5: The study is interesting, but the single, healthy subject used to develop the finite element model without experimental validation limits the impact.

We have added more information on experimental validation (see Methods section, lines 149-154), in which we compared the FE model compressive stiffness to that of a bone-IM nail construct measured experimentally. The reviewer may also wish to see reference 31 for additional details. As for the use of a single subject, we view this aspect as a strength of the study. It would be impossible to accurately experimentally measure the effects of three different IM nails in the exact same TTC specimen, due to the damage caused by implantation and removal of multiple devices. Our methodology allows us to isolate the effects of the IM nails without additional variability introduced by using multiple TTC complex specimens.

Lines 112-113, Page 5: Why not demonstrate the influence of different IM nails with varying material properties, since only one subject was used to develop a single finite element model? In the previous study by the authors, they vary the effect of bone modulus and demonstrate changes. How does the comparison in this manuscript of the different IM nails changes in relation to the bone modulus, and does the positive effect of the dynamization decrease with different bone quality?

These are all good and thoughtful questions, and the authors appreciate your suggestion. Our previous study, which established the pseudoelastic nail model, did quantify the effects of bone quality (in terms of Young’s modulus) on load sharing, and we found that decreasing bone density (and therefore bone modulus) led to a shift in loading from the bone to the nail. Based on those results, we know that the same effect would occur in the other nails analyzed in the current study. As for the effects of bone quality on the effectiveness of dynamization, because the NiTi element is activated and locked into place on the right side of the unloading plateau (see ref. 31 for diagram), there is an additional 3% strain that can be compensated for by contraction of the NiTi element. This is a relatively large amount of “slack” in the system that can be accommodated, and therefore we do not have reason to believe that a change in bone modulus would affect the maintenance of compression in any significant way. While we could analyze an additional set of models with varying bone properties, we feel that it would not fit within the focus of the current study, which aims to understand the effects of nail material properties (the titanium and carbon fiber nails are identical, other than material properties) and dynamization on load sharing and maintenance of compression in the face of localized bone resorption. Additionally, given the reviewer’s suggestion that we should only focus on one aspect (accommodation of resorption), we feel that adding additional models would be contrary to this critique while taking away from the focus of our study.

Lines 129-130, Page 6: Why not demonstrate the influence from different IM nails with different alignment? How sensitive are the results to changes in position and alignment of the device?

All three nails were aligned identically in the TTC complex, as verified by our clinical author (Pacaccio). Fig 5 provides a good visual reference of identical alignment. A parameter study on spatial placement and alignment of the nails would be valuable, but in this study we wanted to focus on comparing the different nail designs and materials. Also, as noted in the comment above, we received a critique stating that we may already be focusing on too many issues.

Lines 132-135, Pages 6-7: Where does the coefficient of friction value come from?

The following literature source is now cited: Yu HY, Cai ZB, Zhou ZR, Zhu MH. Fretting behavior of cortical bone against titanium and its alloy. Wear 2005;259:910-918.

Lines 132-135, Pages 6-7: Is the bone-nail interface perfect? Is that realistic? What implications are there for modeling the IM nails this way? This model representation negates the ability of the device to settle with further loading and potentially change the results. This should probably be mentioned in the Discussion.

The bone-nail interface is “perfect,” because we place the nail in the bone and “overwrite” the bone tissue occupying the same space, producing a space in the bone that is the exact shape of the implant. In a real surgery, a reamer is used to produce a cavity for fitting the nail into place, so the reviewer is correct that the perfect interface is most likely not 100% realistic. However, we do not expect the smooth surface geometry of the interface to have any significant impact on our results. Because we applied contact conditions between the nail and bone, relative motion can occur at the interface, and this should be a reasonable approximation of what happens in a real patient. We have added statements to the discussion to address this issue (lines 397-402). On a closely-related note, we mention at the end of the results section (lines 335-336) and in the limitations portion of the discussion (lines 394-397) that the viscoelastic behavior of bone, which would lead to additional settling, was also not simulated in our models. 

Lines 135-136, Page 7: Do you mean frictionless here? The language used in this sentence does not make this clear.

Thank you for pointing out this misstatement. We revised the text to state that frictionless contact was used in this location.

Lines 142-143, Page 7: I would refer to this as ‘simulated resorption behavior’ because the representation of resorption is simplified.

Thank you for this suggestion. We have made the requested change in the text.

Lines 164-166, Page 8: I am confused by the loading applied. How can you apply this load to just the distal end of the calcaneus where the IM nail was implanted and call it ‘gait loading’? What about during toe-off when the ground reaction force is being applied to the forefoot, which wasn’t modeled? Also, why weren’t the off axis loads applied to assess the complexity of loads during stance? The loading does not seem physiologic if all the load is always applied to the distal calcaneus near the IM nail implanted.

The reviewer is correct that the models do not capture all the complex shifts of loading components and load application areas that occur as a real person takes a step. Including the entirety of the foot and toe anatomy and the change of load application area from the heel to midfoot to toe would capture a more realistic representation of gait, but it would also introduce a whole other level of complexity to what are already a very challenging set of simulations. Our loading scenario is intended to capture the overall shifts in magnitude of the ground reaction force (which are dominated by the vertical force component) that are transmitted through the ankle during a step. This issue is now addressed in the discussion (lines 405-411).

Results

Lines 197-198, Page 9: Change ‘no gait loading’ and ‘peak gait load’ to ‘unloaded’ and ‘peak compressive load’. This manuscript may be basing the compressive force off of the ground reaction forces during gait, but they are only being applied as a range of axial compressive loads. They do not replicate the loads under walking conditions, as these would be more complex.

This change has been made in the text.

Lines 201-210, Page 10: Why focus on reporting absolute values? Since you only have one subject without any experimental validation, the analysis should consist of relative comparisons between the different nails. There is no ability to determine if these values are correct, or that they represent the values that would be demonstrated in others.

We provide the absolute values in order to allow the reader to make direct comparisons between the different models. The use of color maps to visualize the distribution of mechanical loading requires a quantitative number, and we chose von Mises stress as the most appropriate value, since it summarizes the stress state at each location with a single number indicating force concentration that leads to material deformation. As relative differences (for example, percent difference) would still be based on these absolute values, we chose to provide the reader with a direct and straightforward means of comparing the mechanical environments in the models. 

Lines 205-206, Page 10: Why did you compare 400 to 500N? If you can’t achieve 500N with the NiTi nail, then why not just compare each to 400N? I understand that the differences found between the two are vast, even with this discrepancy, but the inconsistency in methodology doesn’t make sense.

These force values correspond to the values that would be applied in clinical application. We wanted to compare the nails at the load values that would actually be used in a patient. The geometry of the NiTi element used in this study limits the force generation capacity to 400 N, while the static nails are loaded at the clinically indicated value of 500 N. During development of the models, we found that a 400 N force in the static nails did not affect our conclusions (due to the vast difference, as the reviewer points out), so we decided to err on the side of clinical accuracy.

Lines 217-224, Pages 10-11: The simulations seem to be quasistatic and not dynamic, so I’m not sure why two cycles of gait loading were performed. It seems like the axial load was just varied. Were dynamic simulations performed and acceleration of loading during walking included?

The reviewer is correct that our models are quasistatic in nature. We used two gait cycles due to the nonlinear behavior of the NiTi compressive element. As described in Figure 3 of ref. 31, the stress generated in the NiTi element is path dependent. It behaves differently under loading and unloading conditions. If you look closely at the NiTi force in Fig. 6 of the current manuscript, you may appreciate some very subtle differences in the two gait cycles. We used two cycles in the original paper (ref. 31) and chose to use the same approach in our current models to ensure that we captured these variations.

Lines 232-234, Page 11: It is not clear where at ‘toe-off’ you are referring to make this calculation.

We refer to the second peak in applied load as “toe off.” This is now stated in the text at the location indicated by the reviewer.

Lines 253-264, Page 12: I like the sensitivity analysis of simulated bone resorption in this study. Why not just focus on this aspect? Since this is one model of a single foot without experimental validation, the sensitivity of different amounts of bone resorption is more interesting and impactful than the earlier analysis.

Thank you for this comment. We worked very hard to develop methods for simulating bone resorption. Other methods were tried, but the thermal contraction used in this study was the most effective. Thank you for recognizing the value of this aspect of the study. We included the analysis of load sharing, because the choice of nail material is an important consideration in device design. Moreover, while the concept of stress shielding is well appreciated in the field of orthopedics, our team has long noticed a dearth of quantitative studies addressing stress shielding. While it is well accepted that stress shielding occurs, as shown by shifting bone density in bones with implants, there is very little information on how much stress shielding occurs. 

Discussion

Lines 310-313, Page 14: Can you really say that this is dynamic compressive loading? It is not clear from the methods that you can. It appears you applied different values of compressive load near the site of the nail.

Thank you for pointing this out. We have changed the text to say “cyclic” compressive loading.

Lines 313-315, Page 14: I would use ‘suggest’ instead of ‘exhibit’ here. The language is a little strong for an analysis with one model.

We now say that the data “suggest a minimal degree of degree of compression…”

Lines 369-372, Page 17: Why not include more specimens then to present results in light of different subject variability in bone material properties, geometry, alignment, etc.?

We focused on a single specimen to isolate the effects of nail design and material. This allowed us to make an “apples to apples” comparison without the added variability introduced by the use of multiple specimens. 

Lines 372-374, Page 17: This is true, but the finite element model would be more impactful with some level of experimental validation and then use this to even extrapolate predictions of stress.

As now stated in the methods section, we did perform an experimental validation of overall construct stiffness, but not site-specific stress or strain values.

---

## [Decision Letter · Decision Letter 1]

15 May 2023

PONE-D-23-03456R1Effect of intramedullary nail stiffness on load-sharing in tibiotalocalcaneal arthrodesis: a patient-specific finite element studyPLOS ONE

Dear Dr. Carpenter,Please insert comments here and delete this placeholder text when finished. Be sure to:

We look forward to receiving your revised manuscript.

Kind regards,

Pawel Klosowski, D.Sc.

Academic Editor

PLOS ONE

Journal Requirements:

Reviewers' comments:

Reviewer's Responses to Questions

**Comments to the Author**

1. If the authors have adequately addressed your comments raised in a previous round of review and you feel that this manuscript is now acceptable for publication, you may indicate that here to bypass the “Comments to the Author” section, enter your conflict of interest statement in the “Confidential to Editor” section, and submit your "Accept" recommendation.

Reviewer #1: All comments have been addressed

Reviewer #2: (No Response)

2. Is the manuscript technically sound, and do the data support the conclusions?

Reviewer #1: Yes

Reviewer #2: Partly

3. Has the statistical analysis been performed appropriately and rigorously? 

Reviewer #1: N/A

Reviewer #2: N/A

4. Have the authors made all data underlying the findings in their manuscript fully available?

Reviewer #1: Yes

Reviewer #2: Yes

5. Is the manuscript presented in an intelligible fashion and written in standard English?

Reviewer #1: Yes

Reviewer #2: Yes

6. Review Comments to the Author

Reviewer #1: (No Response)

Reviewer #2: PONE-D-23-03456.R1

Effect of Intramedullary Nail Stiffness on Load-sharing in Tibiotalocalcaneal Arthrodesis: A Patient-Specific Finite Element Study

Terrill et al.

Comments to Authors:

Response to Author Responses to Reviewers

Although the authors state that their methodology allows them to isolate the effects of the IM nails with their one subject, which is true, it still does not eliminate the need to perform the same analysis in other subject models to support their conclusions. A repeated-measures analysis with multiple subject models does not prevent an investigation to isolate the effects of IM nails. In fact, I think it is important to analysis different aspects that these nails will encounter as a part of a sensitivity analysis in order to say anything valuable about the results. If multiple models generated with different geometry and bone material properties could exhibit the behavior that the authors have stated, then this would improve the impact of their analysis. The problem with this study is that the authors have only found that one subject had the effects they describe, where we don't know whether this is the case with other subjects or with other conditions that are possible like malalignment or reduced bone density. This should at least be addressed further in the limitation section as this is more work, but these details may influence their conclusions.

Introduction

Lines 81-86, Page 4: Include background information and rationale within the earlier paragraphs of the Introduction for evaluating nails with bone resorption modeled. There was no mention of bone resorption until the objective and hypothesis. Please elaborate to prepare the reader for the objective and hypothesis.

Methods

Line 182, Page 9: Change “peak gait load” to “peak compressive load.” You are applying a compressive force from a value extracted from the ground reaction force and applied to the opening of the implanted nail. You are not representing gait loading. Change this throughout the manuscript.

Line 182-184, Page 9: Do you mean “vertical ground reaction force”? Is that where you are getting this value? Same for Lines 187-188.

Results

Lines 242-244, Page 12: Two cycles of gait loading? This is confusing because these are quasistatic simulations. Do you mean that you are running multiple simulations or steps while varying the compressive load applied? If so, clarify as such within the results (even though this is more of a method detail).

7. PLOS authors have the option to publish the peer review history of their article (what does this mean?). If published, this will include your full peer review and any attached files.

Reviewer #1: No

Reviewer #2: No

---

## [Author Response · Author response to Decision Letter 1]

14 Jun 2023

The authors would like to thank the reviewer for their time and effort in thoughtfully evaluating our revised manuscript. Below we have included each question/comment provided by the reviewer along with our responses. The Revised Manuscript with Track Changes document highlights the changes made in response to the critiques, and the changes are described below. Please note that the line numbers listed in our italicized responses below correspond to the line numbers in the marked up copy. We again thank the reviewer for identifying the need for these changes that have resulted in a stronger paper, which we hope is now suitable for publication in PLOS ONE. 

Comments and Responses: 

Response to Author Responses to Reviewers

Although the authors state that their methodology allows them to isolate the effects of the IM nails with their one subject, which is true, it still does not eliminate the need to perform the same analysis in other subject models to support their conclusions. A repeated-measures analysis with multiple subject models does not prevent an investigation to isolate the effects of IM nails. In fact, I think it is important to analysis different aspects that these nails will encounter as a part of a sensitivity analysis in order to say anything valuable about the results. If multiple models generated with different geometry and bone material properties could exhibit the behavior that the authors have stated, then this would improve the impact of their analysis. The problem with this study is that the authors have only found that one subject had the effects they describe, where we don't know whether this is the case with other subjects or with other conditions that are possible like malalignment or reduced bone density. This should at least be addressed further in the limitation section as this is more work, but these details may influence their conclusions.

We agree with the reviewer and do appreciate the value that would be added by analyzing the nail behavior for a range of different subjects’ ankles. We do not have reason to believe that the relative nail behaviors would not be reproduced in other subjects, but we agree that additional analyses would need to be performed in order to quantitatively confirm this. However, given the added complexity in what is already a quite complex set of models (note that we added an 18-page detailed description of the methodology to the supporting information), at this time a comparison across different subjects is out of the scope of the current study. We have addressed this limitation further in the discussion (lines 415-421) and have added a qualifying statement to the conclusions (line 426).

Introduction

Lines 81-86, Page 4: Include background information and rationale within the earlier paragraphs of the Introduction for evaluating nails with bone resorption modeled. There was no mention of bone resorption until the objective and hypothesis. Please elaborate to prepare the reader for the objective and hypothesis.

We have added verbiage and supporting references to raise the issue of localized resorption in the introduction (lines 60-65).

Methods

1. Line 182, Page 9: Change “peak gait load” to “peak compressive load.” You are applying a compressive force from a value extracted from the ground reaction force and applied to the opening of the implanted nail. You are not representing gait loading. Change this throughout the manuscript.

 We have made this change and removed the word “gait” where appropriate (lines 178, 189, 223, 228, 229, 238, 240, 248, 253, 260, 262, and the captions for figures 5, 6, and 7).

2. Line 182-184, Page 9: Do you mean “vertical ground reaction force”? Is that where you are getting this value? Same for Lines 187-188.

 We have changed the wording to “maximum vertical ground reaction forces” in lines 179-180 and “applied load” in line 189.

Results

Lines 242-244, Page 12: Two cycles of gait loading? This is confusing because these are quasistatic simulations. Do you mean that you are running multiple simulations or steps while varying the compressive load applied? If so, clarify as such within the results (even though this is more of a method detail).

For added clarity, we have added an explanation for why we used two cycles (due to the different behavior of NiTi in shortening and lengthening) and have added an explanation of the multiple quasistatic time points (lines 238-244). In order to clarify that the FEA is not in fact dynamic, we also removed the word dynamic from lines 256, 346, 352, 356, 366, and 385, and from the caption for Fig. 6.

---

## [Editor Report · Decision Letter 2]

19 Jun 2023

Effect of intramedullary nail stiffness on load-sharing in tibiotalocalcaneal arthrodesis: a patient-specific finite element study

PONE-D-23-03456R2

Dear Dr. Carpenter,

We’re pleased to inform you that your manuscript has been judged scientifically suitable for publication and will be formally accepted for publication once it meets all outstanding technical requirements.

Kind regards,

Pawel Klosowski, D.Sc.

Academic Editor

PLOS ONE
---

## [Editor Report · Acceptance letter]

21 Jun 2023

PONE-D-23-03456R2 

Effect of intramedullary nail stiffness on load-sharing in tibiotalocalcaneal arthrodesis: a patient-specific finite element study 

Dear Dr. Carpenter:

I'm pleased to inform you that your manuscript has been deemed suitable for publication in PLOS ONE. Congratulations! Your manuscript is now with our production department. 

Kind regards, 

on behalf of

Prof. Pawel Klosowski 

Academic Editor

PLOS ONE